# Effects of Thyroid Powder on Tadpole (*Lithobates catesbeiana*) Metamorphosis and Growth: The Role of Lipid Metabolism and Gut Microbiota

**DOI:** 10.3390/ani14020208

**Published:** 2024-01-08

**Authors:** Bo Zhu, Chuang Shao, Wenjie Xu, Jihong Dai, Guihong Fu, Yi Hu

**Affiliations:** 1Fisheries College, Hunan Agricultural University, Changsha 410128, China; bp.zhu@outlook.com (B.Z.); 13605186516@163.com (C.S.); xwjxqsz@163.com (W.X.); 429022456@hunau.edu.cn (J.D.); snow03221@163.com (G.F.); 2Hunan Engineering Technology Research Center of Featured Aquatic Resources Utilization, Hunan Agricultural University, Changsha 410128, China

**Keywords:** lipid metabolism, gut microbiota, bullfrog, amphibians, metamorphosis

## Abstract

**Simple Summary:**

The production of artificially farmed bullfrogs in China is limited by the low metamorphosis rate of tadpoles. This study investigated the effects of different doses (0 g/kg, 1.5 g/kg, 3 g/kg, 4.5 g/kg, and 6 g/kg) of thyroid powder added to the diets on the metamorphosis and growth of bullfrog tadpoles. The results showed that 4.5 g/kg of thyroid powder could significantly increase the metamorphosis rate and weight gain rate of tadpoles, while adding 6 g/kg of thyroid powder could increase the average weight of tadpoles during metamorphosis. In addition, both doses of thyroid powder could promote the fat breakdown metabolism of tadpoles and alter the intestinal microbiota composition (increasing the abundance of beneficial bacteria, especially *Akkermansia*). In summary, appropriate doses of thyroid powder could promote the metamorphosis and growth of tadpoles, and might be used to solve the problem of the low metamorphosis rate of bullfrogs.

**Abstract:**

A low metamorphosis rate of amphibian larvae, commonly known as tadpoles, limits the farming production of bullfrogs (*Lithobates catesbeiana*). This study aimed to examine the effects of processed thyroid powder as a feed additive on tadpole metamorphosis, lipid metabolism, and gut microbiota. Five groups of tadpoles were fed with diets containing 0 g/kg (TH0), 1.5 g/kg (TH1.5), 3 g/kg (TH3), 4.5 g/kg (TH4.5), and 6 g/kg (TH6) thyroid powder for 70 days. The results showed that TH increased the average weight of tadpoles during metamorphosis, with the TH6 group having the highest values. The TH4.5 group had the highest metamorphosis rate (*p* < 0.05). Biochemical tests and Oil Red O staining showed that the lipid (triglyceride) content in the liver decreased after TH supplementation, especially at doses higher than 1.5 g/kg. RT-qPCR revealed that TH at doses higher than 4.5 g/kg significantly up-regulated the transcriptional expression of the *pparα*, *accb*, *fas*, *fadd6*, *acadl*, and *lcat* genes, which are related to lipid metabolism (*p* < 0.05). These results showed that TH seems to simultaneously promote the synthesis and decomposition of lipid and fatty acids, but ultimately show a decrease in lipids. As for the gut microbiota, it is noteworthy that Verrucomicrobia increased significantly in the TH4.5 and TH6 groups, and the *Akkermansia* (classified as Verrucomicrobia) was the corresponding genus, which is related to lipid metabolism. Specifically, the metabolic pathways of the gut microbiota were mainly enriched in metabolic-related functions (such as lipid metabolism), and there were significant differences in metabolic and immune pathways between the TH4.5 and TH0 groups (*p* < 0.05). In summary, TH may enhance lipid metabolism by modulating the gut microbiota (especially *Akkermansia*), thereby promoting the growth of tadpoles. Consequently, a supplementation of 4.5 g/kg or 6 g/kg of TH is recommended for promoting the metamorphosis and growth of tadpoles.

## 1. Introduction

Bullfrogs (*Lithobates catesbeiana*), native to America, are a fast-growing amphibian species with high nutritional value, used as an important food source in China [1,2]. Tadpoles are the larvae of frogs, and metamorphosis is the process of their transformation into frogs. The current low metamorphosis rate limits the development of the farming of bullfrogs. This is especially true with the shift of the main bullfrog-producing areas to high latitudes in China and falling temperatures. Some bullfrog farms in Hunan province in China were surveyed, and the results showed that the comprehensive metamorphosis rate of tadpoles was only 30~50%, meaning that a low proportion of tadpoles hatched from eggs became froglets. Tadpole metamorphosis is impacted by factors such as temperature [3], the light/dark cycle [4], food availability, and predation [5]. However, there is no effective scientific and economic solution to this problem. Increasing the metamorphosis rate of tadpoles can directly promote industrial development. 

Thyroxine is a key hormone that regulates metamorphosis in tadpoles and acts by binding to nuclear receptors (TRs) that modulate gene expression in various tissues [6,7]. These receptors can activate or repress genes that are involved in metamorphic changes such as cell death, differentiation, and remodeling [8,9]. For instance, they can regulate the death of tail and gill cells and stimulate the differentiation of limb and lung cells [10]. However, the expression of thyroxine is also regulated by temperature, and a lower temperature inhibits thyroxine-induced metamorphosis, which is also a challenge to bullfrog breeding in high latitudes [11,12]. In addition, tadpoles enter metamorphosis only when they accumulate enough nutrients and weight, and a high-quality and sufficient diet could accelerate this process. Thus, promoting the utilization of food nutrients (lipids, proteins, etc.) will accelerate tadpole growth and metamorphosis. It is worth noting that thyroxine has also been shown to regulate lipid metabolism [13]. These facts encourage us to interfere with thyroxine expression or directly supplement thyroxine to promote food digestion and tadpole metamorphosis. 

To address the above problem, this paper investigated whether adding thyroid (TH) powder to the feed could promote the metamorphosis of tadpoles. We studied the effects of TH on the metamorphosis and growth of the bullfrog tadpoles. Our goals were to improve the metamorphosis rate and growth of tadpoles, promote the development of bullfrog breeding industry, and provide enough food for humans.

## 2. Materials and Methods

### 2.1. Diet Preparation

Thyroid powder (TH) is a processed by-product made from pig thyroid tissue by air-drying, low-temperature drying (20 °C), and pulverizing. In the basal diet (Table 1) for tadpoles, 0 g/kg, 1.5 g/kg, 3 g/kg, 4.5 g/kg, and 6 g/kg of TH were added to prepare five diets labeled as TH0, TH1.5, TH3, TH4.5, and TH6, respectively.

The ingredients of each diet were pre-mixed according to the diet formulation and then processed through grinding, blending, pelleting, drying, and grinding to obtain a powder that passed through an 80-mesh sieve. The equipment used in the preparation process was the same as described in our previous publication [14]. The diets were preserved at −20 °C until the end of the feeding trial.

### 2.2. Trial Animals and Feeding Management

The rearing trial was conducted at a farm in Nanning City, Guangxi Zhuang Autonomous Region. After hatching, the tadpoles were acclimated in canvas pools for one week before being used in the trial with an initial weight of approximately 0.03 g. A total of 6000 tadpoles were randomly allocated to 5 groups and reared in 15 different mesh cages. The dimensions of each mesh cage were 0.8 m in length, width, and height, with a tadpole density of 625 per square meter. The tadpoles were fed with five different types of diets. The trial lasted for 70 days. During the rearing period, the tadpoles were fed three times daily (at 7 a.m., 12 p.m., and 5 p.m.) at a rate of 3% to 7% of their body weight per day. The feeding rate was adjusted weekly based on their feed intake and estimated weight. In order to account for the specific feeding behavior of larvae, we maintained a constant amount of food for each group, rather than allowing them free access to food. The average air temperature during the rearing period was 32.9 °C and the average water temperature was 29.5 °C, with dissolved oxygen levels ≥ 4.0 mg/L.

### 2.3. Sample Collection and Analyses

The Gosner stage is a widely accepted classification system for the embryonic and larval development of anuran amphibians, including frogs and toads. There are 46 developmental stages of frog embryos and larvae (tadpoles), with metamorphosis occurring from stage 42 (characterized by the complete development of hind limbs and the emergence of forelimbs) to stage 46 (marked by the complete absorption of the tadpole’s tail) [15].

On days 59 and 70 of the rearing trial, juvenile frogs and tadpoles (with fully developed limbs but incomplete tail resorption, presumably after stage 44, when tadpoles can metamorphose into juvenile frogs with only the tail-retraction stage remaining) were recorded, and their number and weight were recorded to calculate the metamorphosis rates and the average weight of the tadpoles after metamorphosis. On day 70, the number and weight of remaining tadpoles in each cage were also recorded to calculate the feed conversion ratio, weight gain, and survival rate.
(1)$$SR(\%)=\left(\frac{FN}{IN}\right)\times 100$$
(2)$$AW(g)=\left(\frac{FW}{FN}\right)\times 100$$
(3)$$WGR(\%)=\left(\frac{FW}{IW}\right)\times 100$$
(4)$$FCR=\left(\frac{DW}{FW-IW}\right)$$

In the formulations, SR represents the survival ratio, and FN and IN represent the final number and initial number, respectively. AW and FW represent the average weight and final weight, respectively. WG and IW represent the weight gain ratio and initial weight, respectively. FCR and DW represent the feed conversion ratio and weight of diet, respectively. In addition, the abbreviations in this manuscript are described in Table 2.

On day 70, we collected the livers of 36 metamorphic tadpoles from each group and stored them in Eppendorf tubes. We snap-froze the tubes in liquid nitrogen and kept them at −80 °C until analyzing the liver biochemical lipid markers. We measured fatty acid, TG, T-CHO, LDL, TBA, and GLU using reagent kits from Nanjing Jiancheng Bioengineering Institute (Nanjing, China). Before the measurement, we carefully read the operating procedures and precautions, and followed the instructions strictly. 

Another set of 9 liver samples from each group were collected separately, rinsed with saline, and fixed with 4% formalin solution. Paraffin-embedded tissue sections were prepared and stained with hematoxylin and eosin (HE), according to Feldman [16], to observe the liver tissue morphology. Oil red O staining was performed on liver lipid using Marquez’s [17] method to observe liver lipid accumulation. 

An additional set of 27 liver samples from each group were also harvested and frozen to await analysis of the transcriptional expression of genes. We used TRIzol reagent to extract total RNA and assessed its quantity and quality using spectrophotometry and agarose gel electrophoresis, respectively. RT-qPCR was performed to obtain and calculate the gene transcripts of *acadl*, *accb*, *cpt1a*, *cds1*, *fas*, *gk*, *hmgcr*, *lcat*, *ldlr1*, *pemt*, and *pparα*. We used actin beta/gamma 1 (*actb_g1*) as the reference gene and normalized the target genes expression to the *actb_g1* expression. Table 3 lists the primers we used for the quantitative PCR.

The intestinal contents of 18 tadpoles from each group were collected and frozen. DNA extraction and 16S rRNA gene sequencing were performed following the methods described in previous studies [18].

### 2.4. Statistical Analysis

The data were analyzed using SPSS 25.0 software (IBM, New York, NY, USA). We used Shapiro–Wilk and Levene tests to check the normality and homogeneity of data, respectively [19]. For data with normal distribution and homogenous variance, we performed one-way ANOVA and Duncan’s multiple range test. For data with heterogenous variance, we used Welch’s ANOVA and the Games–Howell multiple range test. We set statistical significance at *p* < 0.05 and reported the data as mean ± SEM. 

## 3. Results

### 3.1. Growth and Metamorphosis

The results showed that TH application decreased the feed conversion ratio and increased the total WGR of tadpoles, but the differences were not statistically significant (*p* > 0.05) (Table 4).

TH increased the average weight of tadpoles in the metamorphosis period (AW-IM). The TH6 group showed a significantly higher final body weight than the TH0 group (*p* < 0.05) (Figure 1). The metamorphosis rate and overall metamorphosis rate of tadpoles on day 59 and day 70 reached the maximum in the TH4.5 group (*p* < 0.05) (Figure 2).

### 3.2. Liver Biochemical Tests

The TH3 group had a significantly lower TG content than the TH0 group. The TBA content of all supplement groups was lower than the TH0 group (*p* < 0.05). In addition, all supplement groups had a higher fatty acid content than the TH0 group, and the TH3 and TH4.5 groups had a significantly higher LDL content than the TH0 group (*p* < 0.05) (Figure 3).

### 3.3. Liver HE Staining and Oil Red O Staining

As shown in Figure 4, with the supplementation of TH, the hepatocyte vacuolation tended to decrease, and the vacuolation of the TH4.5 and TH6 groups decreased significantly. As shown in Figure 5, the red staining in liver oil red O sections after thyroid powder supplementation indicated a decreasing trend of neutral lipid content.

### 3.4. Lipid Metabolism-Related Gene Transcription Level 

The transcriptional expression of *pemt* related to lipid metabolism was down-regulated after TH supplementation, while the transcriptional expression of *pparα* and *accb* was up-regulated (*p* < 0.05). The TH supplementation also up-regulated the expression of *cpt1a* and *gk* in the TH3 group (*p* < 0.05) and the expression of *fas*, *fadd6*, *acadl*, and *lcat* in the TH4.5 and TH6 groups (*p* < 0.05) (Figure 6).

### 3.5. Gut Microbiota

The results of gut microbiota (Figure 7) show that the chao1 and observed species indices in each group decreased after TH supplementation. In addition, except for the TH group, the Shannon and Simpson indices also decreased. These showed that TH reduced the abundance and diversity of gut microbiota. 

The composition of gut microbiota in Figure 8a shows that the Chloroflexi decreased in each supplement group, while Verrucomicrobia was increased in the TH4.5 and TH6 groups at the phylum level. From the composition of the genus level shown in Figure 8b, the *Caldilinea* (classified as Chloroflexi) decreased, while *Cetobacterium* increased. It is worth noting that the genus *Akkermansia* (classified as Verrucomicrobia) was increased in the TH4.5 and TH6 groups. Further Lefse analysis identified *Akkermansia* as a biomarker to distinguish the 4.5 g/kg group from the other groups (LDA = 3) (Figure 9).

The potential metabolic pathways of gut microbiota are mainly enriched in metabolic functions (lipid metabolism, carbohydrate metabolism, etc.) (Figure 10a). The main differential metabolic pathways between the TH4.5 and TH0 groups were ko00196 (Photosynthesis-antenna proteins), ko00531 (Glycosaminoglycan degradation), ko00331 (Clavulanic acid biosynthesis), and ko04621 (NOD-like receptor signaling pathway) (Figure 10b). 

## 4. Discussion

The results showed that 4.5 g/kg of TH significantly accelerated metamorphosis, which is consistent with our hypothesis. This may be mediated by thyroxine, which is a vital substance for initiating and regulating the metamorphosis process [20]. Metamorphosis is triggered by a threshold level of thyroxine that activates the expression of thyroxine receptors (TRs) and their target genes [21]. The expression and secretion of thyroxine peak during the metamorphic climax and return to normal levels after metamorphosis is completed [22,23]. Thyroxine regulates programmed cell death and proliferation and differentiation in various tissues during metamorphosis via different molecular pathways. For instance, thyroxine triggers apoptosis in the tail, gills, and fins through caspases, Bcl-2 family members, and p53, while it stimulates cell cycle progression and differentiation in the limbs, lungs, and skin by activating cyclins, CDKs, and FGFs [6,10]. Thyroxine also affects epigenetic modifications such as DNA methylation during metamorphosis, which influence the functions of various genes [24,25]. In addition, there is a report indicating that tadpoles can respond to exogenous thyroxine before their own thyroid tissue develops fully [20]. Therefore, we conclude that the tadpole metamorphosis in this trial is influenced by TH in diets, in which thyroxine regulates various physiological changes and promotes the transformation of tadpoles to frogs.

In addition to increasing the metamorphosis rate, TH significantly increased the weight gain of tadpoles during metamorphosis, which also seems to be attributable to thyroxine. Firstly, thyroxine could stimulate growth factors and interact with IGF and GH to promote body growth [26]. Moreover, thyroxine is a vital regulator of animal metabolism, as it affects the function and activity of various organs and cells (such as cardiomyocytes and reproductive cells) involved in energy production and utilization, which can influence body weight [27,28]. Consistently, we found that TH enhanced the transcriptional expression of genes related to lipid synthesis and breakdown, which resulted in a decrease in neutral lipids (biochemical analysis, oil red O staining). Triglyceride is the main storage form of lipids in animals, consisting of three fatty acids attached to a glycerol molecule [29]. Triglyceride and fatty acids can be interconverted by enzymatic reactions. Thyroxine promotes lipid breakdown and the β-oxidation of fatty acids to release energy, which may contribute to the enhanced growth of tadpoles in the high-TH supplementation group. Recent research has revealed that thyroxine induces hepatic triglyceride catabolism by activating autophagy, a process that degrades lipid droplets and delivers fatty acids to mitochondria for β-oxidation [13]. Thyroxine has also been found to regulate the gene expression involved in fatty acid metabolism, such as CPT1A and ACOX, which encode enzymes or proteins that facilitate fatty acid transport into mitochondria, the oxidation of fatty acids, and the dissipation of energy as heat [30]. Further, some diiodothyronines, which are products of thyroxine catabolism, have also been shown to have metabolic effects in humans [31], such as modulating mitochondrial function and energy expenditure [32]. In short, TH could promote animal growth by increasing the metabolism of nutrients. 

As mentioned above, although this trial showed a reduction in neutral lipids and an increase in fatty acids, the gene expression involved in triglyceride synthesis and fatty acid degradation was also up-regulated. Thus, a possible explanation is that TH could simultaneously promote lipid decomposition and synthesis, but triglyceride breakdown is greater than synthesis [33], and fatty acid production is greater than breakdown. The effects of thyroxine on lipid decomposition have been investigated previously, and we have also found the effects of thyroxine regulating lipid production from some studies. Previous studies have reported that thyroxine affects fatty acid uptake by stimulating lipolysis from dietary fat sources [34]. Thyroxine also increases the gene expression of fatty acid uptake, such as CD36 and FATP4, which encode proteins that mediate the fatty acid transport across the plasma membrane into hepatocytes for lipid synthesis [33]. In addition, thyroxine directly affects hepatic triglyceride anabolism by regulating the gene expression involved in lipogenesis, such as ACC, FAS, SREBP-1c, and SCD-1 [35], and also regulates these processes by influencing the lipogenic gene transcription and the activities of enzymes such as LPL and HL [36]. 

Gut microbiota have been shown to regulate lipid metabolism in animals [37]. Medium-chain fatty acids, short-chain fatty acids, and bile acids produced by gut microbiota from diet can affect lipid metabolism and inflammation via the PPARα, NF-κB, and MAPK signaling pathways or binding some receptors [38,39,40]. Thus, the state of gut microbiota could reflect the metabolism and health of animals [41]. This study found that the function of gut microbiota is mainly enriched in metabolic-related pathways and related to lipid metabolism. Consistently, the abundance of *Akkermansia* in the TH4.5 and TH6 groups was significantly higher than in the TH0 group, especially in the TH4.5 group. *Akkermansia* bacteria are biomarkers that distinguish them from the TH0 group. We speculate that *Akkermansia* bacteria are important microbiota affecting lipid metabolism. In fact, *Akkermansia* belongs to the Verrucomicrobia phylum, is mainly colonized in the mucous layer of the intestine, and has been found to alleviate metabolic disorders and obesity in animals [42]. *Akkermansia* can modulate lipid metabolism by enhancing intestinal barrier function, increasing fatty acid oxidation and energy expenditure in adipose tissue [42,43], and regulating bile acid metabolism and signaling in the intestine and liver [44]. In addition, the NOD-like receptor (NLR) signaling pathway was also significantly activated in the TH4.5 group, besides the significant difference in the abundance of *Akkermansia* bacteria from the TH0 group. The NLR is a pathway that mediates the innate immune response to intracellular pathogens and cellular stress, and activates downstream signaling cascades [45]. For instance, it can induce the production of pro-inflammatory cytokines by forming multiprotein complexes called inflammasomes, activate NF-κB and MAPK pathways that regulate gene expression related to inflammation, and modulate autophagy [46]. *Akkermansia* has also been found to improve immune function through the NLR signaling pathway, such as enhancing the intestinal barrier function by increasing mucus production and tight junction expression, and reducing inflammation and oxidative stress by modulating NLRP3 inflammasome activation and IL-1β secretion [47,48]. Nevertheless, the reason for the increase in *Akkermansia* abundance after adding TH cannot be well explained. 

## 5. Conclusions

In summary, TH supplementation can promote the metamorphosis of bullfrog tadpoles and enhance lipid and fatty acid metabolism, while gut microbiota, especially *Akkermansia*, play a vital role in affecting lipid metabolism and potentially also immune function. Thus, it is advisable to supplement 4.5 g/kg or 6 g/kg thyroid powder (TH) in the diet of tadpoles, which can contribute to the development of the bullfrog breeding industry and provide more food for humans.

## 6. Perspectives

In China, the artificial breeding industry of bullfrogs faces an important problem, which is the low metamorphosis rate of tadpoles. In recent years, the production area of bullfrogs has shifted to the northern high-latitude regions, where the environmental temperature is lower and unfavorable for tadpole metamorphosis. Therefore, it is necessary to find ways to promote tadpole metamorphosis, which is not only applicable to bullfrog tadpoles, but also to other artificially bred frogs, such as *Pelophylax nigromaculatus* and *Quasipaa spinosa*. This study found that thyroid powder can accelerate the metamorphosis of bullfrog tadpoles in the same feeding cycle, and has potential application value. However, the safety of thyroid powder (its impact on animals and the environment) still needs to be evaluated through more research. In addition, a feasible method is to add thyroid hormone promoters (such as iodine and tyrosine) to the feed, which may indirectly promote metamorphosis by increasing the synthesis of the thyroid hormone. This method is more acceptable, because these substances are already present in commercial feed, and only the dosage would need to be adjusted.

## Figures and Tables

**Figure 1 animals-14-00208-f001:**
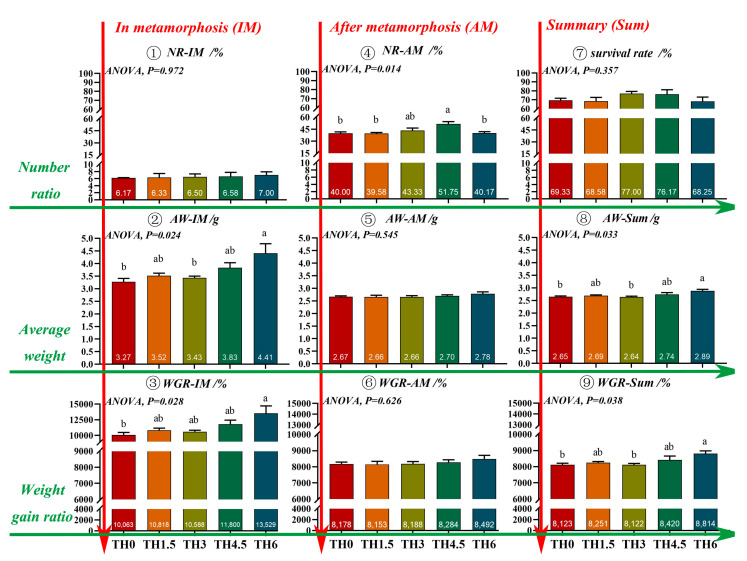
Number ratio, average weight, and weight gain ratio of the tadpoles at different developmental periods on day 70. NR-IM (number ratio of tadpoles in metamorphosis), AW-IM (average weight of tadpoles in metamorphosis), WGR-IM (weight gain ratio of tadpoles in metamorphosis), NR-AM (number ratio of tadpoles after metamorphosis), AW-AM (average weight of tadpoles after metamorphosis), WGR-AM (weight gain ratio of tadpoles after metamorphosis), AW-Sum (average weight of all tadpoles), and WGR-Sum (weight gain ratio of all tadpoles). Here, the average weight of tadpoles was used to calculate the weight gain ratio. Groups possessing the same letter indicate that they are not significantly different (*p* > 0.05).

**Figure 2 animals-14-00208-f002:**
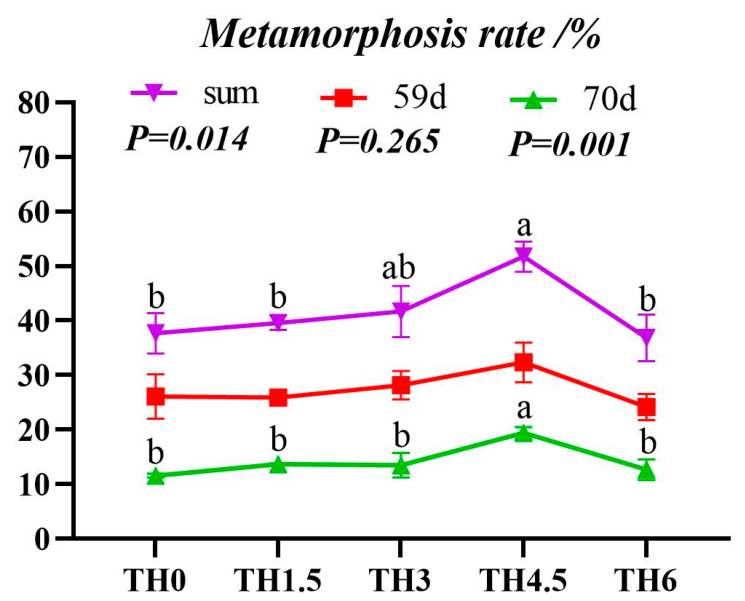
Tadpole stage metamorphosis rate. The red line is the metamorphosis rate on day 59, the green line is the metamorphosis rate on day 60 to day 70, and the purple line is the total metamorphosis rate. Groups having the same letter in the same line indicate no significant difference (*p* > 0.05).

**Figure 3 animals-14-00208-f003:**
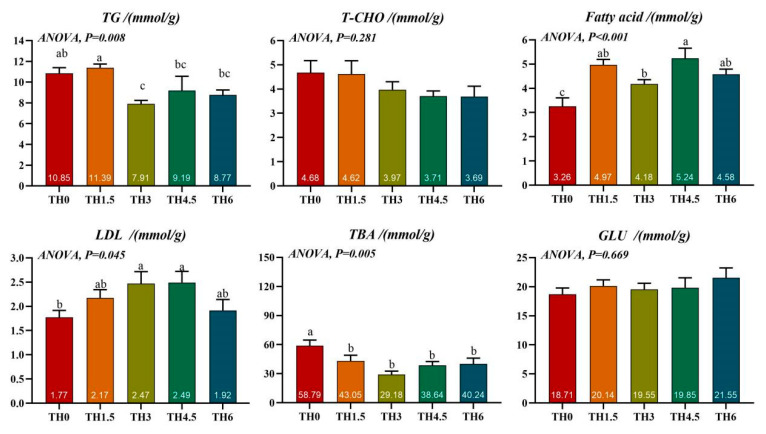
Liver lipid-related biochemical indicators. TG (triglyceride), T-CHO (total cholesterol), LDL (low-density lipoprotein), TBA (total bile acid), and GLU (glucose). Groups possessing the same letter indicate that they are not significantly different (*p* > 0.05).

**Figure 4 animals-14-00208-f004:**
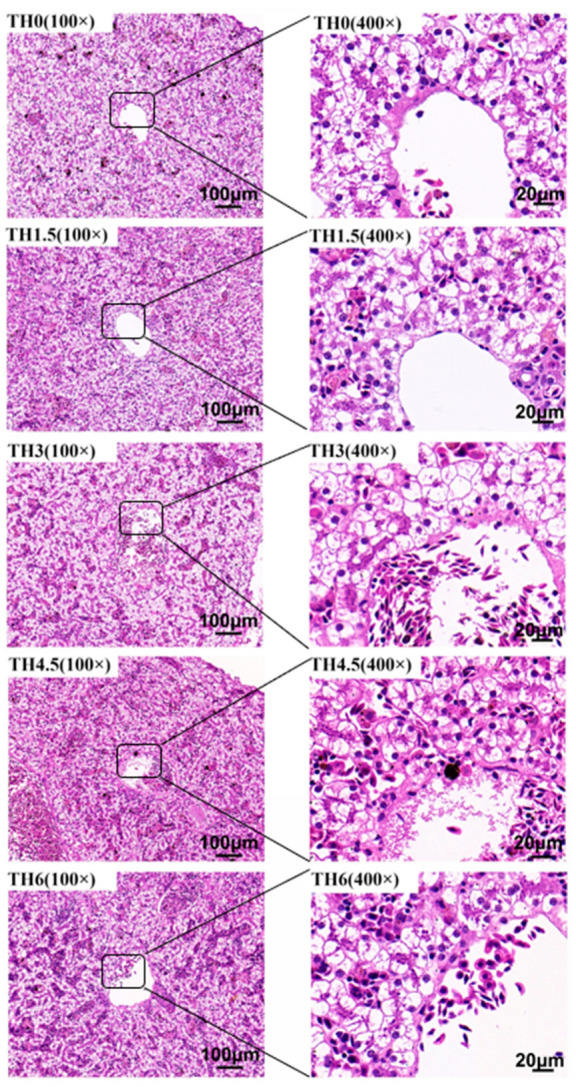
Liver hematoxylin-eosin (HE) staining sections. The 100× indicates that the magnification of the optical microscope is 100, and 400× indicates that the magnification is 400.

**Figure 5 animals-14-00208-f005:**
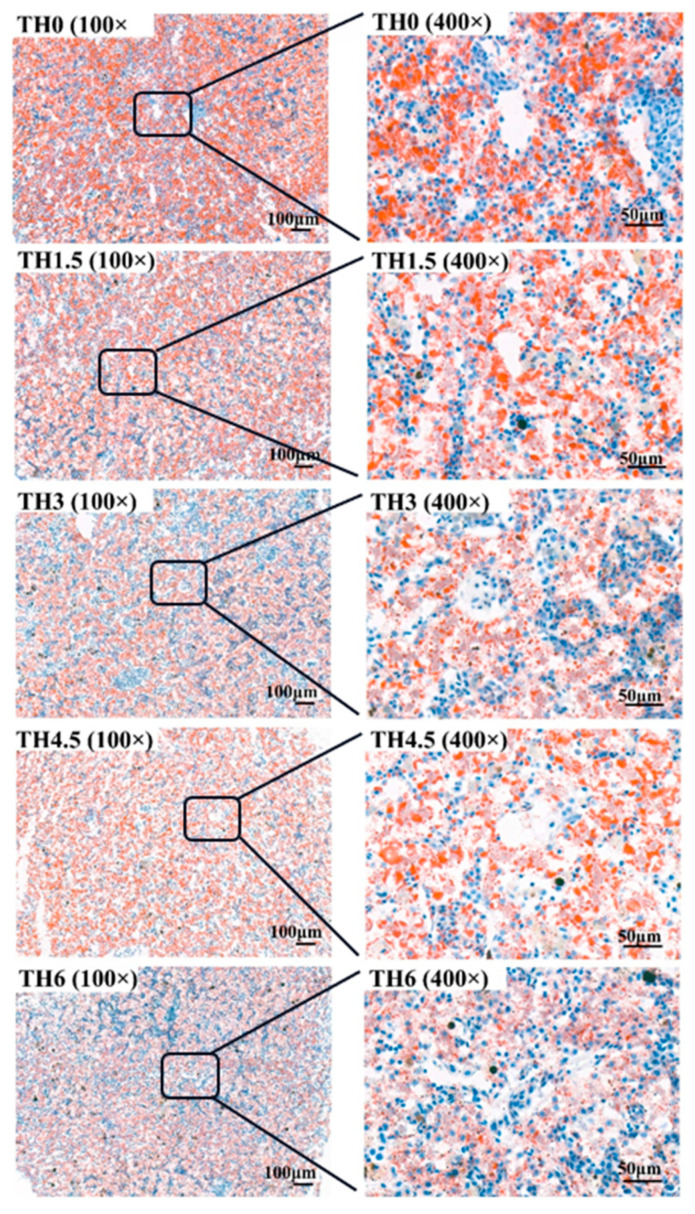
Liver oil red O staining sections. The 100× indicates that the magnification of the optical microscope is 100, and 400× indicates that the magnification is 400.

**Figure 6 animals-14-00208-f006:**
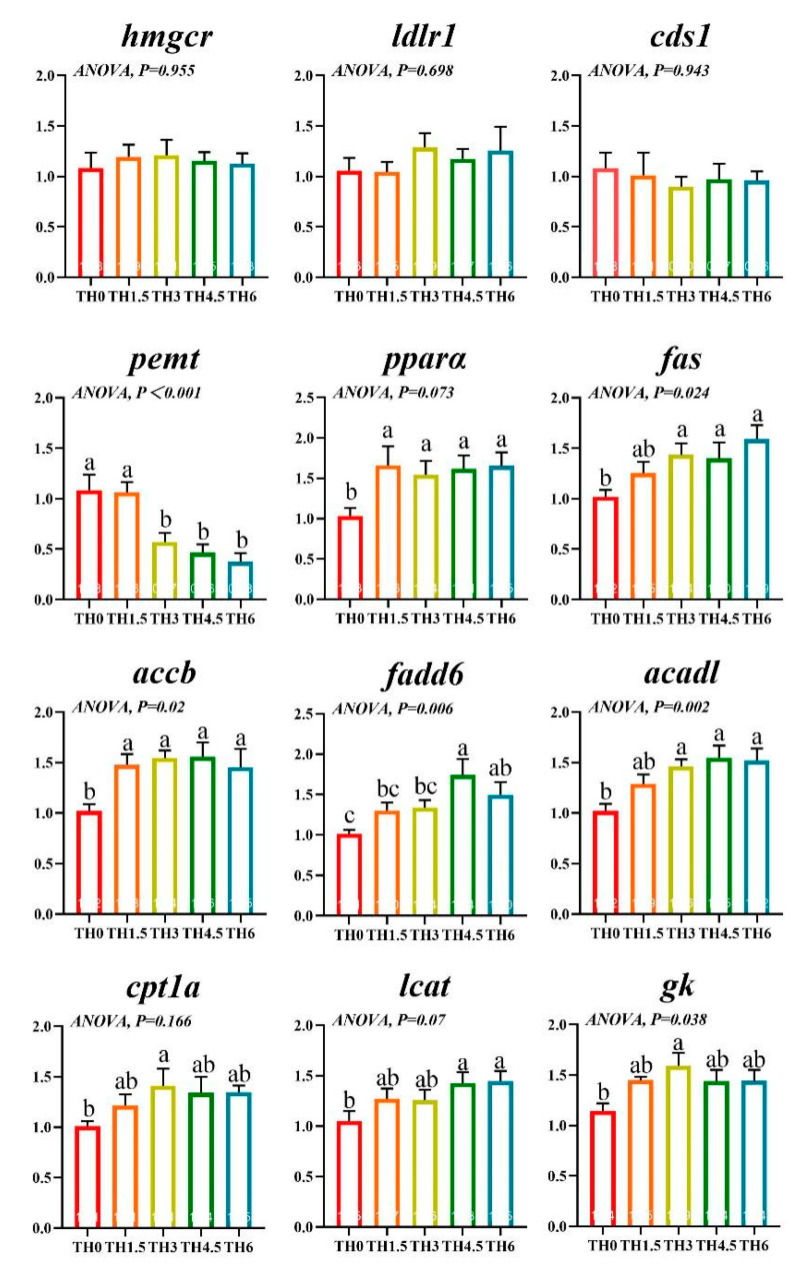
Transcription level of genes related to liver lipid metabolism. *hmgcr* (3-hydroxy-3-methylglutaryl-CoA reductase), *ldlr1* (low-density lipoprotein receptor 1), *cds1* (CDP-diacylglycerol synthase), *pemt* (phosphatidylethanolamine N-methyltransferase), *pparα* (peroxisome proliferator-activated receptor alpha), *fas* (fatty acid synthase), *accb* (acetyl-CoA carboxylase beta), *fadd6* (fatty acid desaturase delta-6)*, acadl* (long-chain-acyl-CoA dehydrogenase), *cpt1a* (carnitine palmitoyltransferase 1a), *lcat* (lecithin-cholesterol acyltransferase), and *gk* (glycerol kinase). Groups possessing the same letter indicate that they are not significantly different (*p* > 0.05).

**Figure 7 animals-14-00208-f007:**
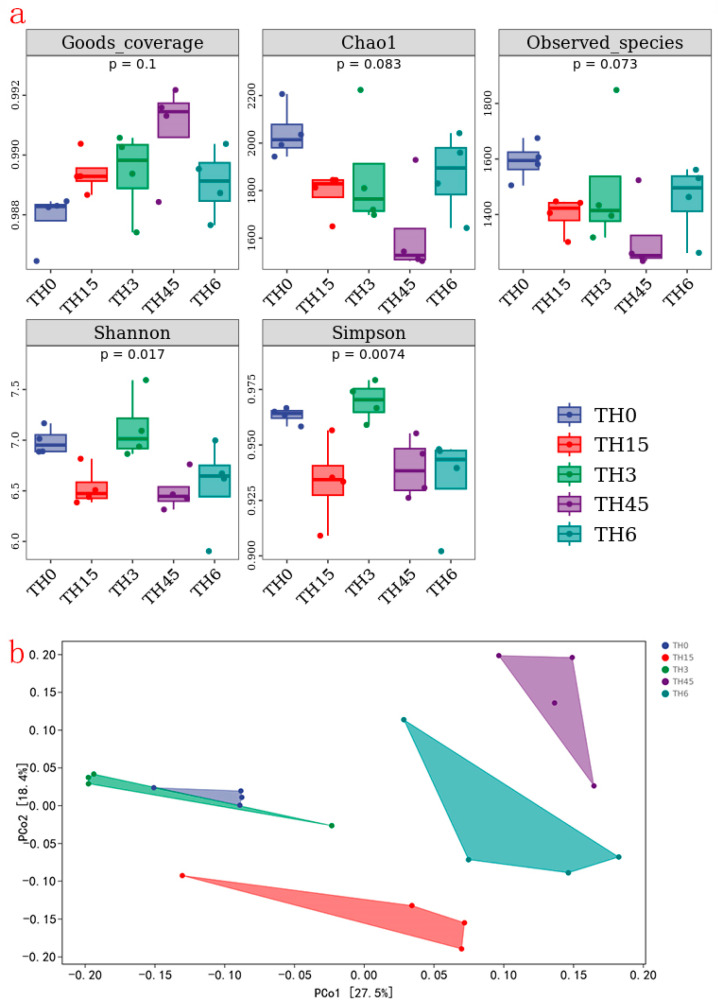
The α diversity index and β diversity index of gut microbiota. (**a**) α diversity index (contains goods_coverage, chao1, observed_species, shannon, simpson indices). (**b**) β diversity index (principal co-ordinates analysis of gut microbiota).

**Figure 8 animals-14-00208-f008:**
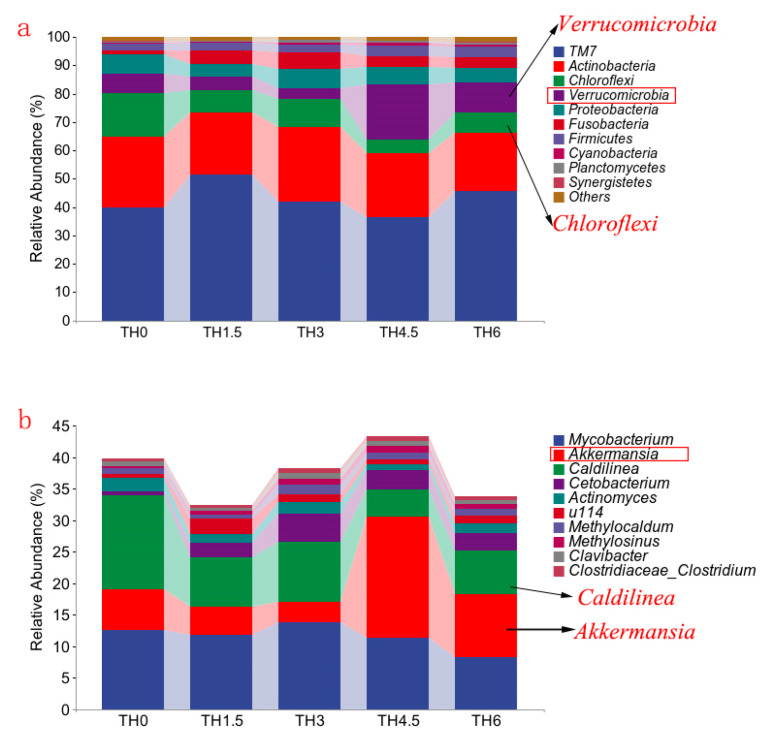
Composition of the gut microbiota (phylum and genus levels). (**a**) Column chart of the top 10 species in relative abundance (phylum level). (**b**) Column chart of the top 10 species in relative abundance (genus level).

**Figure 9 animals-14-00208-f009:**
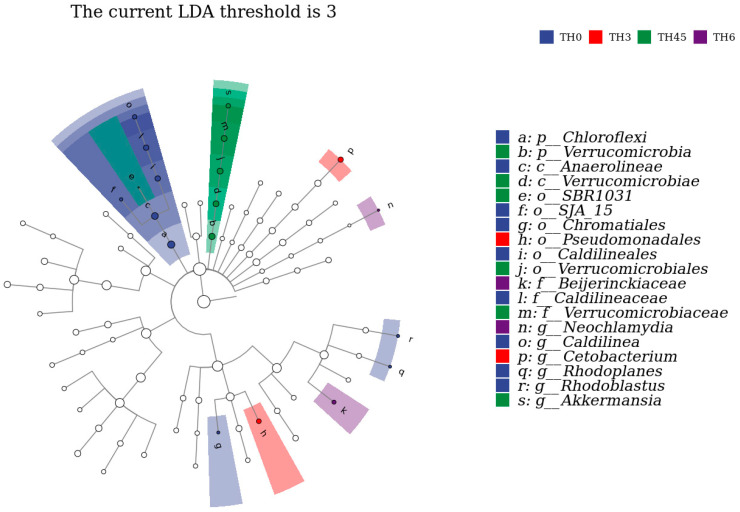
LDA effect size analysis and the clade map of the gut microbiota. The rings represent species, genus, family, order, class, and phylum from outside to inside. The species with an LDA SCORE > 3 were defined as statistically different biomarkers.

**Figure 10 animals-14-00208-f010:**
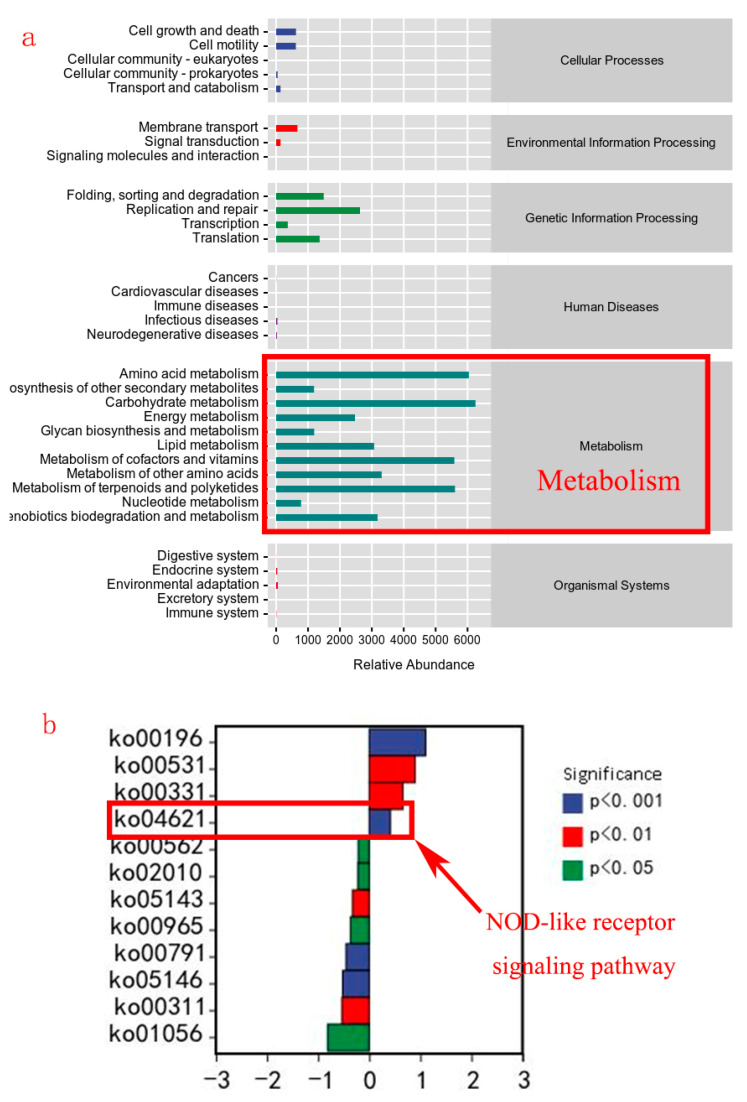
The difference in predicted functional metabolisms in gut bacterial Kyoto Encyclopedia of Genes and Genomes (KEGG) pathways. (**a**) The main enriched functional pathways. (**b**) Differential metabolic pathways in the 0 g/kg group and the 4.5 g/kg group.

**Table 1 animals-14-00208-t001:** Formulation and chemical composition of the diet (% air dry basis).

Item	TH0	TH1.5	TH3	TH4.5	TH6
Fish meal	15.00	15.00	15.00	15.00	15.00
Chicken powder	20.00	20.00	20.00	20.00	20.00
Soybean meal	10.00	10.00	10.00	10.00	10.00
Soy protein concentrate	18.00	18.00	18.00	18.00	18.00
Rice bran	8.00	8.00	8.00	8.00	8.00
Wheat flour	23.95	23.80	23.65	23.50	23.35
Thyroid powder (TH)	0	0.15	0.30	0.45	0.60
Soybean oil	1.00	1.00	1.00	1.00	1.00
Vitamin and mineral premixes ^※^	2.00	2.00	2.00	2.00	2.00
Ca(H_2_PO_4_)_2_	1.50	1.50	1.50	1.50	1.50
Choline chloride	0.50	0.50	0.50	0.50	0.50
Ethoxyquin	0.02	0.02	0.02	0.02	0.02
Mold inhibitor	0.03	0.03	0.03	0.03	0.03
Proximate composition					
Dry matter	92.31	92.31	92.31	92.31	92.31
Crude protein	43.42	43.42	43.42	43.42	43.42
Crude lipid	6.68	6.68	6.68	6.68	6.68
Ash	7.58	7.58	7.58	7.58	7.58

Note: ^※^ One kilogram of premix contains: vitamin A 900,000 IU, vitamin D3 170,000 IU, vitamin E 7500 mg, vitamin K3 960 mg, vitamin B1 1200 mg, vitamin B2 1800 mg, vitamin B6 1200 mg, vitamin B12 6.4 mg, vitamin C 15,600 mg, D-calcium pantothenate 4500 mg, niacin 9000 mg, folic acid 520 mg, D-biotin 30 mg, inositol 7000 mg, magnesium 4500 mg, zinc 3500 mg, manganese 2100 mg, copper 880 mg, iron 5200 mg, cobalt 160 mg, iodine 100 mg, selenium 30 mg, and so on. Rice bran was used to fill up to 1 kg.

**Table 2 animals-14-00208-t002:** List of abbreviations.

Abbreviations	Explanation
*acadl*	acetyl-CoA carboxylase beta
*accb*	acetyl-CoA carboxylase beta
*actb_g1*	actin beta/gamma 1
*cpt1a*	carnitine palmitoyltransferase 1a
*cds1*	CDP-diacylglycerol synthase
*fas*	fatty acid synthase
*gk*	glycerol kinase
*hmgcr*	3-hydroxy-3-methylglutaryl-CoA reductase
*lcat*	lecithin-cholesterol acyltransferase
*ldlr1*	low-density lipoprotein receptor 1
*pemt*	phosphatidylethanolamine N-methyltransferase
*fadd6*	fatty acid desaturase delta-6
*pparα*	peroxisome proliferator-activated receptor alpha
TG	triglyceride
T-CHO	total cholesterol
LDL	low-density lipoprotein
TBA	total bile acid
GLU	glucose
TH	thyroid powder

**Table 3 animals-14-00208-t003:** List of primers used in the real-time quantitative PCR.

Gene	Sequence, 5′→3′	GenBank Number
*acadl*	Fwd: TGAGGAAACCCGGAACTATGTC Rev: TGTGCTGCACGGTCTGTAAGT	LH364687.1
*accb*	Fwd: GTTAAAGCTGCCATCCTCACTGT Rev: TGTCCGTCTGGCTAAGATGGT	LH212450.1
*actb_g1*	Fwd: ATGATGCTCCTCGTGCTGTGT Rev: CCCCATTCCAACCATGACA	LH355272.1
*cpt1a*	Fwd: TGATTGGCAAAATCAAAGAACATC Rev: AATGCTCTGACCCTGGTGAGA	LH022414.1
*cds1*	Fwd: GGTTTCTGCATGTTTGTGTTGAG Rev: TCCATCCAAACATGTAAAACTGAAG	LH114672.1
*fas*	Fwd: CCTCCACGCCAGAACAAGAT Rev: GATATTTTTATGAGTGGACATTGTATCGA	LH228595.1
*gk*	Fwd: AACGCTTTGAGCCACAGATTAAT Rev: CTGCTTTTTTCCATCGAGCAT	LH193866.1
*hmgcr*	Fwd: TGCATCCTCAAAAACCCAGAT Rev: GGGATGTGTTTAGCATTCACCAA	LH363056.1
*lcat*	Fwd: GCTGTAGGGTGACCTGTTCCAT Rev: AGATACGAAGGGCCTTCTGGAT	LH171224.1
*ldlr1*	Fwd: AAGGCTACCAACTAGATCCAGTAACTG Rev: CGGTTGGTGAAGAACAGGTATG	LH243159.1
*pemt*	Fwd: CCGATATACGGTGACCCAAAA Rev: ACCCGCTCTTCTGGAATGTG	LH373164.1
*fadd6*	Fwd: TGGATCCTTGCTGAATATGTTAGG	LH144230.1
	Rev: AAGGGAGCTTCAGCCAACTG	
*pparα*	Fwd: CCCGACATTCGATGTTTAGAGATT Rev: CCAGCCCATCTTCTATCACCTT	LH193621.1

**Table 4 animals-14-00208-t004:** Effects of thyroid powder on the growth of bullfrog tadpoles.

	Initial Average Weight/g	Initial Total Weight/g	Final Total Weight/g	Total Weight Gain Ratio/%	Feed Conversion Ratio
TH0	0.03 ± 0.00	12.88 ± 0.03	733.92 ± 15.65	5597.26 ± 128.36	1.18 ± 0.03
TH1.5	0.03 ± 0.00	12.87 ± 0.06	738.13 ± 51.23	5630.70 ± 372.67	1.18 ± 0.08
TH3	0.03 ± 0.00	12.83 ± 0.03	811.91 ± 15.87	6227.31 ± 137.21	1.06 ± 0.02
TH4.5	0.03 ± 0.00	12.87 ± 0.03	834.37 ± 54.48	6384.41 ± 437.83	1.04 ± 0.07
TH6	0.03 ± 0.00	12.95 ± 0.05	785.69 ± 37.73	5967.86 ± 301.60	1.10 ± 0.05

Note: here, the total weight of tadpoles was used to calculate the total weight gain and feed conversion rate.

## Data Availability

The data of this study are available from the corresponding author upon reasonable request.

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
