# Peer review of "Effects of Thyroid Powder on Tadpole (Lithobates catesbeiana) Metamorphosis and Growth: The Role of Lipid Metabolism and Gut Microbiota"

_animals, 2024, doi:10.3390/ani14020208_

Round 1
Reviewer 1 Report
Comments and Suggestions for Authors
Review MDPI Animals thyroid for metamorphosis
Paper is a bit in the middle of scientific and applied to frog farmers.
- On one side you lack a universal unit of measurement of your extract so no other research team can repeat your experiment
- On the other side, a frog breeder/farmer will not be able to use your protocol because “kitchen” recipe is not detailed.
Keywords: add metamorphosis and amphibians or tadpoles
Introduction
Line 38: add frogs to precise “their transformation into frogs”
Line 38: the word artificial breeding induces confusion as it often refers to reproduction processes. Sentence is understood as if thyroid would improve reproduction processes. Which is not this papers purpose. Better use the word farming and remove artificial.
Line 38: as you base your paper on frog farms results, you should source your statement: low metamorphosis rate. And give some average rates observed in frog farms/hatcheries.
Line 44: you MUST precise that thyroxine is an hormon.
Line 57 is part of M&M
Materials and Methods
Line 64: basal diet. Please explain that it is (or not) the current diet / proximate composition used by the farm in Nanning City.
Line 65: you MUST explain how you processed the porcine gland to obtain the powder. And refer to is as home-made powder, as no brand can be citated here.
Line 66: please make a DOSAGE of your extract to express thyroid supplementation in a value that can be repeated by other research teams and confronted by other future papers.
Line 66: why 6 is the maximum? On which basis did you decide to use 6 and not 20 or 100
Line 83: all papers dealing with fish/tadpole rearing always mention initial density in individuals/liter AND the mode of water treatment: RAS or water change. This strongly influences and must be presented.
Line 95: you should introduce here Gosner table for metamorphosis stages.
Results
Line 141: you might want to precise : decreased (thus improved) the feed conversion.
Line 148 table 4: if total weight was used, the please express the initial weight in total also. It makes this table confuse : animals growing from 0,03 to 5000!
Line 152 figure 1: graph 1, 4, 7: it is not a number rate, but a ratio expressed in %. Please adapt legend both in graph and in Figure description.
Line 152: in each period on the day 70. I’m sorry it is not clear. You earlier mention 0 to 59 and 59 to 70. Where is the third period?
Line 152 figure 1: why are first and second periods equal in all conditions? I suppose weight gain and survival rate differ already at day 59. Or did I totally misunderstand this part?
Line 159: Figure 2: it would be informative to add the survival rate for the 5 conditions.
Line 160: “on day 70” isn’t it “from day 60 to day 70”?
Line 214, figure 8 : totally impossible to read. Please improve
Discussion
Line 241: “as early as in the development”. Please be more rigorous: do you mean early larval development? Which Gosner stage?
Line 243: metamorphosis is regulated: verb is confusing as many other parameters regulate metamorphosis. You might want to say improved of influenced
Line 304: affect is both way. Should you write improve
Conclusions
Line 313 same comment as in M&M: unit must be more scientific and universal
Line 314: promote is a bit too strong, as 20% increase in metamorphosis rate indeed is an improvement, but will not alone promote bullfrog industry. You might want to use the verb contribute. The economical consideration is NOT a conclusion of your work. It is rather a perspectives, and I think you should add a Perspectives paragraph. Also, by suggesting the addition of thyroxin into tadpole diet you should at least mention the legal veterinary and sanitary context, at least in your country: would that be authorised?
Perspectives (to be added)
Here you might speak about frog farming industry and write about economical value of your work for frog farmers.
Would be interesting to conclude on perspectives, for example at which stage should any thyroxine extract be added in the larvae cycle? Could thyroxine promovers be used instead? As iodin, or aminoacids?
Bibliography
Gosner metamorphosis table should be citated .
Or Nieuwkoop et Faber 1956 : Nieuwkoop P. D. et Faber J. (1956). Normal Table of Xenopus laevis (Daudin). Elsevier/North-Holland, New York
Brown D. D. et Cai L. (2007), Amphibian metamorphosis. Dev. Biol. 306 (1) : 20 - 33 should be citated as it well explains metamorphosis stages and hormons influence along tadpole development and metamorphosis.
Reviewer 2 Report
Comments and Suggestions for Authors
This study investigated the effects of thyroid (TH) powder on metamorphosis and growth of tadpoles. This trial was well conducted.
Line 36, please add the reference. Where was the data from?
Line 64-65, please add the source of thyroid powder, or how to process it?
Porcine thyroid powder was added in diets, how to balance the formula? Table 1, please present the formula of all the diets.
Table 1, in the title, it should be air dry basis, but not dry matter.
Line 83, please add the size of cage.
The information in Fig 1, I suggest convert to table and merge with table 1. After all, value could be more precise than figure.
Table 4, it should be Feed conversion ratio, but not Feed conversion rate.
In the section of 2.4, how many tadpoles were sampled for measurements?
In the REFERENCE, some titles were italic, some were not. Please unify them.
Comments on the Quality of English Languageminor revision
Round 2
Reviewer 2 Report
Comments and Suggestions for Authors
Line 36, please add the reference. Where was the data from?
Table 1, the TH inclusion has been added from 0 to 0.6%, but what ingredient inclusion level was reduced? Otherwise, the total would exceed 100%.
Comments on the Quality of English Language
Minor revision.
Author Response
Thank you again for your comments and suggestions on our manuscript.
1. Regarding the first question "Line 36, please add the reference. Where was the data from?". We answered it in our first reply. It is that this data is recorded on the official website of the Chinese government and in the press releases issued by the official media. And we attached some relevant web links in our reply, but the sources are described in Chinese characters, so they may not be suitable to be cited in papers that may be published in this journal, and since they are cited from web pages rather than academic papers, they are not directly cited in the manuscript. In order to avoid this problem, we have modified this part in the new revised version, i.e., explicit numerical expressions have been deleted, as reflected in the latest version of the manuscript.
2. Regarding the second question, "Table 1, the TH inclusion has been added from 0 to 0.6%, but what ingredient inclusion level was reduced? Otherwise, the total would exceed 100%". Thank you for pointing this out, there were some errors in Table 1, that is, after we added the thyroid powder, accordingly the amount of flour was reduced to keep the total of the recipe at 100%, which we have corrected in the new version of the manuscript.
Once again, we express our heartfelt thanks to you.